# How Epinephrine Administration Interval Impacts the Outcomes of Resuscitation during Adult Cardiac Arrest: A Systematic Review and Meta-Analysis

**DOI:** 10.3390/jcm12020481

**Published:** 2023-01-06

**Authors:** Wachira Wongtanasarasin, Karan Srisurapanont, Daniel K. Nishijima

**Affiliations:** 1Department of Emergency Medicine, Faculty of Medicine, Chiang Mai University, Chiang Mai 50200, Thailand; 2Department of Emergency Medicine, UC Davis School of Medicine, Sacramento, CA 95817, USA

**Keywords:** cardiopulmonary resuscitation, epinephrine, interval, cardiac arrest

## Abstract

Current guidelines for treating cardiac arrest recommend administering 1 mg of epinephrine every 3–5 min. However, this interval is based solely on expert opinion. We aimed to investigate the impact of the epinephrine administration interval (EAI) on resuscitation outcomes in adults with cardiac arrest. We systematically reviewed the PubMed, EMBASE, and Scopus databases. We included studies comparing different EAIs in adult cardiac arrest patients with reported neurological outcomes. Pooled estimates were calculated using the IVhet meta-analysis, and the heterogeneities were assessed using Q and *I^2^* statistics. We evaluated the study risk of bias and overall quality using validated bias assessment tools. Three studies were included. All were classified as “good quality” studies. Only two reported the primary outcome. Compared with a recommended EAI of 3–5 min, a favorable neurological outcome was not significantly different in patients with the other frequencies: for <3 min, odds ratio (OR) 1.93 (95% CI: 0.82–4.54); for >5 min, OR 1.01 (95% CI: 0.55–1.87). For survival to hospital discharge, administering epinephrine for less than 3 min was not associated with a good outcome (OR 1.66, 95% CI: 0.89–3.10). Moreover, EAI of >5 min did not pose a benefit (OR 0.87, 95% CI: 0.68–1.11). Our review showed that EAI during CPR was not associated with better hospital outcomes. Further clinical trials are necessary to determine the optimal dosing interval for epinephrine in adults with cardiac arrest.

## 1. Introduction

Cardiopulmonary arrest (CPA), both out of hospital (OHCA) and in hospital (IHCA), is a leading cause of morbidity and mortality worldwide [1]. Despite significant resources and efforts allocated to improve patient outcomes, overall survival to hospital discharge from OHCA is less than 10% [2,3].

Epinephrine has been recognized as the pharmaceutical cornerstone of cardiopulmonary resuscitation (CPR) for decades because it has a positive effect during CPR by constricting arteries and arterioles via α-adrenergic receptors [4,5]. Its action rapidly enhances coronary artery perfusion pressure, which is important for achieving a return of spontaneous circulation (ROSC) and may contribute to increased survival [6,7]. Current resuscitation algorithms by the American Heart Association (AHA) suggest administering 1 mg of epinephrine every 3–5 min [8]. However, conflicting data support this dosing interval [9,10]. To our knowledge, no large, randomized trials have investigated how frequently epinephrine should be given during adult CPA. Despite the epinephrine recommendation by current guidelines [8], the debate on the benefit of routine administration of epinephrine during CPA is rising. A previous meta-analysis found that giving epinephrine during traumatic OCHA might not improve survival [11]. Two studies reported that higher epinephrine administration intervals (EAIs) might be associated with worse outcomes [12,13]. In contrast, a study by Warren et al. demonstrated that longer average intervals between epinephrine doses were associated with improved survival outcomes [9]. The objective of our study was to conduct a systematic review and meta-analysis to evaluate the impact of EAI on resuscitation outcomes during adult CPA.

## 2. Materials and Methods

This systematic review was prepared and reported according to the PRISMA Extension Statement for Reporting of Systematic Reviews Incorporating Network Meta-analyses of Health Care Interventions [14]. This study’s protocol was prospectively registered on the PROSPERO website (registration ID: CRD42022337212).

### 2.1. Search Strategy and Study Selection

Without language restrictions, we conducted a database search from their inceptions to 30 June 2022, including PubMed, Embase, and Scopus. We also searched for citations from relevant articles. The Medical Subject Heading terms were a combination of search terms with different spellings and endings: “epinephrine,” “adrenaline,” “interval,” “administration,” “dosing interval,” “cardiac arrest,” and “heart arrest”. We obtained the search results retrieved from these databases and removed duplicates. Articles that were not duplicated were imported into the Rayyan QCRI website, and their abstracts were independently checked and evaluated by two authors (W.W. and K.S.). A consensus discussion was used to resolve any discrepancies.

### 2.2. Inclusion Criteria and Outcome of Interest

The selection criteria were as follows:Any study including adults 18 years of age and older who received CPR.At least one arm reported EAI during CPA of 3–5 min.Comparing EAIs other than 3–5 min (higher or lower intervals).Reporting of an outcome on neurological status at hospital discharge.

We included randomized clinical trials and prospective and retrospective observational studies. We excluded preclinical studies, review articles, and studies without a control group (e.g., case reports and case series). Favorable neurological outcome was defined by a cerebral performance category of 1–2 or a modified Rankin scale of 0–3. Two authors (W.W. and K.S.) independently screened the search results to identify eligible studies. Two authors assessed the retrieved studies’ full-text articles against the prespecified criteria. Any disagreements were discussed with a third party and concluded by consensus.

The primary outcome was a favorable neurological outcome at hospital discharge. We selected it as a primary outcome since it is the actual patient-oriented outcome that can explain why we apply this method to CPA patients. Furthermore, this outcome is recommended by consensus guidelines as the outcome measure. Secondary outcomes included the rates of ROSC, survival to hospital admission, and survival to hospital discharge.

### 2.3. Data Extraction and Assessment of the Study Risk of Bias

We developed and piloted a data collection form to collect (1) first author; (2) publication year; (3) type of study; (4) study location, setting, and study enrollment period; (5) number, sex, and age of participants; (6) initial presenting cardiac rhythms; (7) witnessed and bystander CPA; (8) definitions of intervention and comparisons; and (9) outcomes of interest. We attempted to contact the corresponding author by email for missing or incomplete data in an original publication or clarification of the data.

Each study’s risk of bias was independently assessed by two authors (W.W. and K.S.) using the Newcastle—Ottawa scale (NOS) for assessing the quality of observational trials [15] and the Cochrane risk of bias for randomized controlled trials (RCTs) [16]. Any differences were handled through a discussion or referred to a third person who was an expert in this field for a final decision.

### 2.4. Data Synthesis and Statistical Analysis

Relevant information was collected and entered into the Microsoft Excel worksheet. We calculated the odds ratios (ORs) and their 95% confidence intervals (CIs) of the outcome difference between each group. We evaluated studies for clinical heterogeneity before including them in the pooled analysis. The recently developed IVhet model, an estimator under the fixed-effect model assumption with a quasi-likelihood-based variance structure, was used to pool data since it is more appropriate for evaluating meta-analysis with a high heterogeneity [17]. The IVhet model is considered more reasonable than the traditionally used random-effects and fixed-effects models [17]. The Q and I^2^ statistics were also calculated to assess heterogeneity and inconsistency in the data. Values of less than 25%, 25–50%, and more than 50% were classified as low, moderate, and high heterogeneity, respectively [18]. Visual assessment of funnel plots and Egger’s test were planned to visually assess publication bias, which may arise from small-study effects, but was not created due to the inclusion of only three studies [19]. All analyses were conducted using the MetaXL software [20]. All tests were two-tailed, with a *p*-value of <0.05 considered statistically significant.

## 3. Results

### 3.1. Characteristics and Quality of the Included Studies

We identified 1433 relevant citations (Figure 1). After duplicate removal and initial screening, 42 articles were evaluated for eligibility. Two studies were included after the full-text assessment, and one was identified from citation searching and included in this review.

Included studies were published between 2014 and 2021 and enrolled 47,783 participants. All included studies met eligibility for pooled analysis [9,13,21]. Two trials included OHCA patients, whereas the other included IHCA patients. Two were conducted in North America, and the other in Japan. Participants ranged from 68 to 76 years old, and most were male. Witnessed arrest and bystander CPA varied among articles. Two articles reported the primary outcome, while all reported the outcome of survival to hospital discharge. Only one study reported the remaining secondary outcomes, including the rate of ROSC and survival to hospital admission. Table 1 summarizes the characteristics and details of the included studies. Since all included articles were observational studies, we evaluated the study risk of bias using the NOS checklist. All articles were classified as “good quality” studies (Table 2).

### 3.2. Favorable Neurological Outcome at Hospital Discharge

Two studies (*n* = 26,874) recorded favorable neurological status at discharge [13,21]. Based on the highly heterogeneous data (I^2^ = 95% and 87%, for comparing standard interval (3–5 min) with <3 min and >5 min, respectively), patients with standard EAI during CPA might not demonstrate a benefit for favorable neurological outcome at hospital discharge (OR 1.93, 95% CI 0.82–4.54 and OR 1.01, 95% CI 0.55–1.87 for comparing standard interval with <3 min and >5 min, respectively, Figure 2). Furthermore, Egger’s test could not be assessed due to the small number of trials.

### 3.3. Survival to Hospital Discharge

All included studies reported outcomes on survival to hospital discharge [9,13,21]. In one study, we applied the 1-month survival as the survival to hospital discharge outcome in this review [21]. Based on the observed heterogeneous data, administering epinephrine for less than 3 min was not associated with improved survival to hospital discharge (OR 1.66, 95% CI 0.89–3.10, I^2^ = 97%, Figure 3a). Moreover, giving epinephrine longer than 5 min did not show a benefit (OR 0.87, 95% CI 0.68–1.11, I^2^ = 80%, Figure 3b).

## 4. Discussion

Our review summarizes the evidence on EAI during adult CPA and resuscitation outcomes. Although we could not find any RCTs throughout our searches, all included observational studies were classified adequately. Our meta-analysis showed that EAIs shorter (<3 min) or longer (>5 min) than the current EAI recommendations (3–5 min) were not correlated with an improved favorable neurological status at hospital discharge.

For decades, epinephrine, also known as adrenaline, has been the mainstay for the treatment of CPA [6,8]. It benefits the CPA because it improves coronary and cerebral perfusion pressures (CPPs) via artery and arteriole constrictions regulated by α-adrenergic receptors [22,23]. However, nonspecific vasoconstriction may deteriorate postresuscitation outcomes [22]. An animal study found that epinephrine may improve CPP during CPA, but resulted in negative effects during the postresuscitation period [22]. The current recommendations for EAI (1 mg every 3 to 5 min) are based on only expert opinion without supporting evidence [8]. Previous studies have attempted to address how epinephrine impacts neurologically intact survival [5,6,11]. Literature on epinephrine in adult patients has mostly focused on the impact of total or accumulative epinephrine doses on resuscitation outcomes [24,25]. Pharmacological studies indicate that giving epinephrine more frequently than recommended by guidelines may be beneficial [26,27]. Retrospective clinical studies have supported more and less frequent epinephrine administration in pediatric and adult CPA [4,9,12,13,21,26]. Interestingly, the total epinephrine dose during resuscitation was related to a poorer neurological outcome [24,25]. Recently, the landmark PARAMEDIC-2 trial compared the standard amount and administration interval of epinephrine with a placebo toward survival at hospital discharge [6]. Although epinephrine caused a significant increase in ROSC and a modest survival benefit, it was associated with many survivors with impaired neurological status [6]. Our findings are consistent in that more frequent dosing might be associated with increased survival in IHCA patients; however, a frequent alternative interval than usual was not correlated with a positive outcome on neurologically intact survival.

Previous studies of adult and pediatric CPAs have explored the relationship between average EAIs and outcomes; all found that longer average dosing intervals were linked with better results [9,10,12]. The discrepancy between these articles and our findings might be multifaceted. First, our review included both OHCAs and IHCAs, while all these three studies included only IHCAs, which may have unique characteristics. Second, epinephrine during IHCA was delivered mostly in the first 2 to 4 min of the arrest, which may influence the optimum dosage interval. Interestingly, two studies demonstrated that a short administration interval enhanced survival, but these effects were reversed after adjusting for other potential confounders, suggesting a complicated epinephrine-timing connection [9,10].

This review underlines that EAI has been a debatable issue for CPA resuscitation since its inception [28]. Although the current EAI recommendations have not been scientifically investigated, we found that this recommended interval suggested by the American Heart Association was not associated with an improved favorable neurological status. Our findings support that CPA should be tailored to each patient’s needs [29,30,31]. A one-size-fits-all strategy may not be applied to all CPA situations. In addition, our study may be considered hypothesis generation and serve as a basis for future research.

Our results should be interpreted in the context of several limitations. First, we identified only three observational studies. These studies did not consistently document the dosing intervals during CPA. In addition, unrecognized confounders potentially influenced the results of the studies. Second, we observed high statistical heterogeneity in our results. Such heterogeneity might be caused by the differences in other treatments and how each study adjusts for potential confounding variables. Indeed, we conducted the inverse variance heterogeneity analysis to limit statistical errors that may have arisen from the high heterogeneity. Although we did not specifically select the population of OHCA or IHCA patients, no differences were detected in a subgroup analysis. Meta-regression could be conducted to find some factors that can explain the difference between groups; however, it is recommended that meta-regression should only be considered when at least ten studies are in a meta-analysis [32]. Furthermore, as we mentioned earlier, one included article revealed that a short delivery interval improved survival. Still, these results were reversed after controlling for other possible confounders, implying a complicated epinephrine–timing relationship. The authors mentioned that when they removed the CPA duration variable from their regression model, the pattern of associations between EAI and survival was similar to that of the unadjusted model. This review may add another observation to the ongoing debate about the role of epinephrine and EAI in CPA. A prospective, well-designed study is warranted to determine the appropriate dose frequency of epinephrine in treating CPA.

## 5. Conclusions

Our review demonstrated that EAI during adult CPA might not be associated with neurologically favorable survival. This study emphasizes that RCTs are needed and open the door to scientific discussion before discussing the current guidelines. Future randomized clinical trials are required to determine the optional dosing interval of epinephrine in adults with CPA.

## Figures and Tables

**Figure 1 jcm-12-00481-f001:**
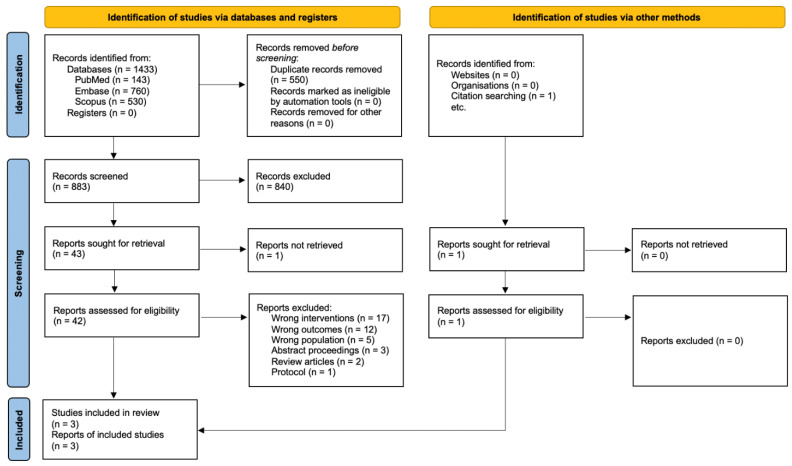
The PRISMA flow diagram.

**Figure 2 jcm-12-00481-f002:**
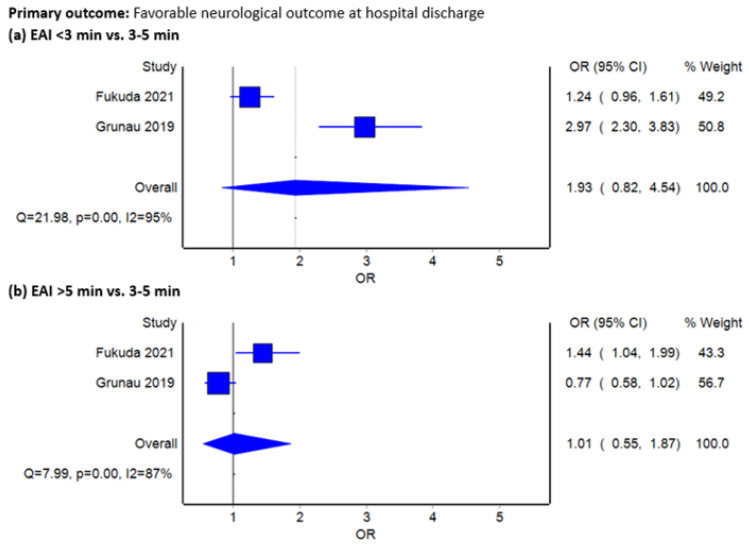
Forest plots comparing favorable neurological status at hospital discharge between standard epinephrine administration interval (every 3–5 min) and (**a**) <3 min or (**b**) >5 min using inverse variance heterogeneity model.

**Figure 3 jcm-12-00481-f003:**
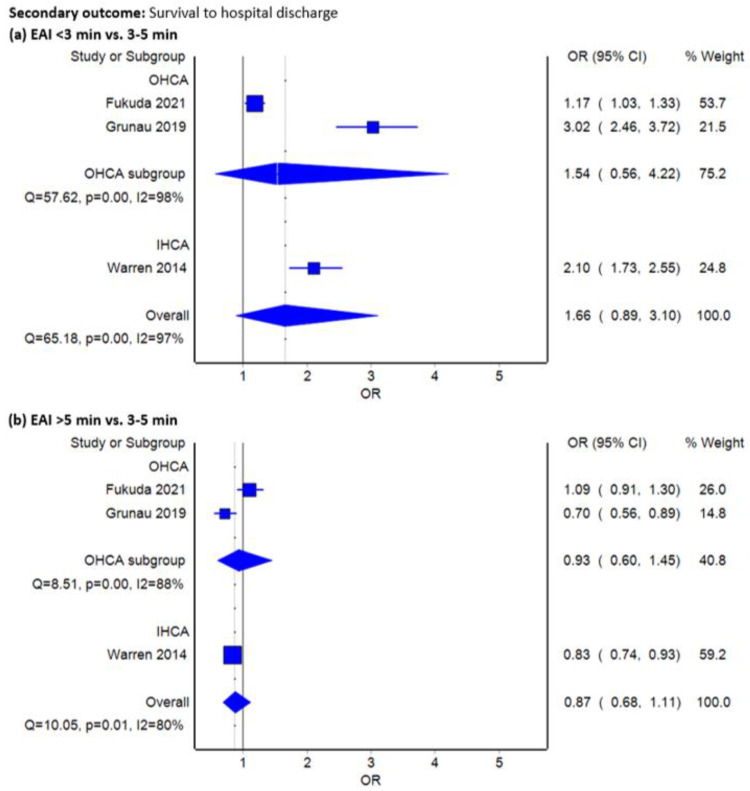
Forest plots comparing survival to hospital discharge between standard epinephrine administration interval (every 3–5 min) and (**a**) <3 min or (**b**) >5 min using inverse variance heterogeneity model.

**Table 1 jcm-12-00481-t001:** Characteristics of included studies.

Study (Publication Year)	Type of Study, Study Location, Enrollment Period	Participants (Total Enrolled; Mean Age ± SD)	Epinephrine Administration Interval	Number of Patients	Male, %	Age, Years (Mean ± SD)	Shockable Rhythm, %	Witnessed Arrest, %	Bystander CPR, %	Presumed Cardiac Cause, %
Fukuda et al. (2021) [21]	Retrospective observational study, Japan, 2011–2017	Adult OHCAs (*n* = 10,965), 75.8 ± 14.3	<3 min	3446	2165, 62.8	73.0 ± 15.1	551, 16.0	2312, 67.1	1877, 54.5	1939, 56.3
			3–5 min	5995	3518, 58.7	77.2 ± 13.6	615, 10.2	4024, 67.1	3162, 52.7	3403, 56.8
			>5 min	1524	879, 57.7	76.3 ± 14.5	205, 13.5	1054, 69.2	866, 56.8	853, 56.0
Grunau et al. (2019) [13]	Retrospective observational study, USA and Canada, 2011–2015	Adult OHCAs (*n* = 15,909), 68 (56–80) ^a^	<3 min	2059	1287, 62.5	68 (56–81) ^a^	413, 20.1	805, 40.2	960, 47.3	N/A
			3–5 min	8599	5604, 65.2	68 (56–80) ^a^	1527, 17.8	3184, 37.0	3998, 46.5	N/A
			>5 min	5251	3397, 64.7	67 (55–79) ^a^	1062, 20.2	2074, 40.3	2226, 42.9	N/A
Warren et al. (2014) [9]	Retrospective observational study, USA, 2000–2009	Adult IHCAs (*n* = 20,909), 68 ± 16	<3 min	1100	674, 61.3	67.0 ± 16.1	152, 13.8	779, 70.8	N/A	N/A
			3–5 min	6093	3720, 61.1	67.6 ± 15.9	709, 11.6	4042, 66.3	N/A	N/A
			>5 min	13,716	8304, 60.5	67.5 ± 15.8	1744, 12.7	9021, 65.8	N/A	N/A

^a^ Reported as the median (interquartile range). Abbreviations: CPR: cardiopulmonary resuscitation; IHCA: in-hospital cardiac arrest; NA: not applicable; OHCA: out-of-hospital cardiac arrest; SD: standard deviation.

**Table 2 jcm-12-00481-t002:** Quality assessment using the Newcastle-Ottawa scale of included studies.

Study	Selection				Comparability ^†^	Outcome			Total Score (Out of 9)
	Representativeness of Exposed Cohort (Maximum: ✩)	Selection of Non-Exposed Cohort (Maximum: ✩)	Ascertainment of Exposure (Maximum: ✩)	Demonstration That Outcome of Interest Was Not Present at Start of Study (Maximum: ✩)	Comparability of Cohorts Based on the Design or Analysis (Maximum: ✩✩)	Assessment of Outcome (Maximum: ✩)	Follow-Up Length (Maximum: ✩)	Loss to Follow-Up Rate (Maximum: ✩)	
Fukuda et al., 2021 [14]	✩✩	✩	✩	✩	✩✩	✩	✩	✩	9
Grunau et al., 2019 [13]	✩	✩	✩	✩	✩✩	✩	✩	✩	9
Warren et al., 2014 [9]	✩	✩	✩	✩	✩✩	✩	✩	✩	9

^†^ Scores were allocated for the primary outcome; one point was given if the study adjusted for initial presenting rhythm, with an additional point given if adjusted for initial presenting rhythm, age, and total cardiopulmonary resuscitation duration.

## Data Availability

Data are available upon requirement.

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
