# Peer review of "How Epinephrine Administration Interval Impacts the Outcomes of Resuscitation during Adult Cardiac Arrest: A Systematic Review and Meta-Analysis"

_jcm, 2023, doi:10.3390/jcm12020481_

Round 1

Reviewer 1 Report

the hypothesis of the study is valid, but the quality of the studies considered is low, as the adjustment variables in a case with cardiac arrest are very many and standardizing the studies or for a meta-analysis is very very difficult, I would say unlikely to be able to obtain useful information from a clinical point of view. Given the methodological limitations of the studies, I believe it is unlikely that the reader will gain any useful information from them.

Author Response

Response to Reviewer 1

We would like to thank the editor and reviewers for their helpful suggestions. We have already revised the manuscript and hope that the revised manuscript has met the requirements for publication. Changes were marked with track changes function in the MS Word.

Comment: the hypothesis of the study is valid, but the quality of the studies considered is low, as the adjustment variables in a case with cardiac arrest are very many and standardizing the studies or for a meta-analysis is very very difficult, I would say unlikely to be able to obtain useful information from a clinical point of view. Given the methodological limitations of the studies, I believe it is unlikely that the reader will gain any useful information from them.

Response: Thank you for your comment. It is important to carefully consider the quality and limitations of the studies used in a meta-analysis. If the studies being included have significant restrictions or biases, it can affect the overall validity and reliability of the findings. Despite the number of included studies being only three; however, all were graded “low risk of bias” using the Newcastle-Ottawa Scale. In the case you describe, if the adjustment variables in cardiac arrest studies are numerous and difficult to standardize, it may be challenging to draw reliable conclusions from a meta-analysis or systematic review of these studies. It may be necessary to consider other types of evidence or research designs to understand the topic comprehensively. We have discussed these points in the ‘Discussion’ section, especially the last paragraph, which is the limitations of this study. (Page 7, Line 342-361)

Reviewer 2 Report

 Authors propose an interesting systematic review and meta-analysis to compare epinephrine administration interval (EAI) below 3 minutes and more than 5 minutes vs EAI of 3-5 minutes in cardiac arrest.

Results are that “EAI 2-3 minutes during CPR was not associated with increased neurologically intact survival. More frequent EAI than recommended by guidelines may be associated with improved survival to hospital discharge.”

The major concern is that only 3 studies were included in the study and all of them are observational retrospective studies. Moreover, the I2 score is very high because of high heterogeneity between studies. Any strong conclusion cannot be proposed.

Because of this limitation and absence of RCTs found during the screening phase included in the meta-analysis the international guidelines must be respected even they come from experts’ consensus. As discussed by authors, more RCTs are needed.   

To examine the influence of population characteristics on overall heterogeneity, authors should use the IVhet model in the Microsoft Excel® MA package, designed particularly for use in meta-analyses with high heterogeneity (MetaXL, available at www.epigear.com). This method uses the quasi-likelihood estimator as an alternative to random-effects models with the problem of underestimation of the statistical error and overconfident estimates.

Another concern that authors should discuss is to have combined OHCA and IHCA. A Meta-regression could be done combining the three studies and find some factors that can explain the heterogeneity. But separation OHCA and IHCA could not bring relevant results because of the lack of studies.

Otherwise, there are no more concerns to address.

Author Response

Response to Reviewer 2

We would like to thank the editor and reviewers for their helpful suggestions. We have already revised the manuscript and hope that the revised manuscript has met the requirements for publication. Changes were marked with track changes function in the MS Word.

Comment: Authors propose an interesting systematic review and meta-analysis to compare epinephrine administration interval (EAI) below 3 minutes and more than 5 minutes vs EAI of 3-5 minutes in cardiac arrest.

Results are that “EAI 2-3 minutes during CPR was not associated with increased neurologically intact survival. More frequent EAI than recommended by guidelines may be associated with improved survival to hospital discharge.”

The major concern is that only 3 studies were included in the study and all of them are observational retrospective studies. Moreover, the I2 score is very high because of high heterogeneity between studies. Any strong conclusion cannot be proposed.

Response: Thank you for your comments. We agreed that the major concern of this review is that only three observational and retrospective studies were included. Also, high heterogeneity was observed, as indicated by the high I2 score. Such a strong conclusion should not be stated. However, we proposed that the results from this review could be considered a hypothesis-generation for future research. We have revised the conclusion in the revised manuscript. It now reads, “Our review demonstrated that EAI during adult CPA might not be associated with neurologically favorable survival. Future randomized clinical trials are needed to determine the optional dosing interval of epinephrine in adults with CPA.” (Page 8, Line 415-417)

Comment: Because of this limitation and absence of RCTs found during the screening phase included in the meta-analysis the international guidelines must be respected even they come from experts’ consensus. As discussed by authors, more RCTs are needed.   

Response: We agree with the reviewer. Results from this review may not directly be implied in real-world clinical settings concerning the study’s limitations. However, we believe this review may add another observation to the ongoing debate about the role of epinephrine and EAI in CPA. More RCTs on this issue are needed. (Page 7, Line 358-361)

Comment: To examine the influence of population characteristics on overall heterogeneity, authors should use the IVhet model in the Microsoft Excel® MA package, designed particularly for use in meta-analyses with high heterogeneity (MetaXL, available at www.epigear.com). This method uses the quasi-likelihood estimator as an alternative to random-effects models with the problem of underestimation of the statistical error and overconfident estimates.

Response: Thank you for giving us this interesting point. The previous article demonstrated that using the IVhet model, which is an estimator under the fixed effect model assumption with a quasi-likelihood-based variance structure, can address the known problems of underestimation of the statistical error and falsely high estimates of confidence with the random-effect model [1]. We have re-performed all analyses using IVhet function in MetaXL [2] and revised the “2.4 Data synthesis and statistical analysis” subsection in the Methods, “3.2 Favorable neurological outcome at hospital discharge” subsection and “3.3 Survival to hospital discharge” in the Results, and figures 2-3 in the revised manuscript. (Page 3, Line 158-172, Page 4, Line 218-232, and Figures 2-3)

Comment: Another concern that authors should discuss is to have combined OHCA and IHCA. A Meta-regression could be done combining the three studies and find some factors that can explain the heterogeneity. But separation OHCA and IHCA could not bring relevant results because of the lack of studies.

Response: Thank you for pointing out this issue. We agreed that meta-regression could be done to find some factors that can explain the difference between groups; however, it is recommended that meta-regression should only be considered when at least ten studies are included in a meta-analysis [3]. We have addressed this point in the limitation section of the revised manuscript. (Page 7, Line 350-353)

References

  1. Doi, S.A.R.; Barendregt, J.J.; Khan, S.; Thalib, L.; Williams, G.M. Advances in the Meta-Analysis of Heterogeneous Clinical Trials I: The Inverse Variance Heterogeneity Model. Contemp Clin Trials 2015, 45, 130–138, doi:10.1016/j.cct.2015.05.009.
  2. EpiGear International MetaXL 2016.
  3. Thompson, S.G.; Higgins, J.P.T. How Should Meta-Regression Analyses Be Undertaken and Interpreted? Stat Med 2002, 21, 1559–1573, doi:10.1002/sim.1187.

Round 2

Reviewer 2 Report

Authors took into account all of my suggestions. The major concern is the very low number of studies for the meta analysis which explain the high heterogeneity. The results can be considered as negative. There is no impact on outcomes either below 3 minutes or over 5 minutes. This study emphasize that RCTs are needed and open the door to scientific discussion before discussing the current guidelines. The English be evaluated by a native English speaking.

Author Response

jcm-2060582: Response to Reviewer 2 (Round 2)

We want to thank the editor and reviewers for providing feedback and comments. We have revised the manuscript according to those comments and hope that the revised manuscript has met the criteria for publication. We provided point-by-point responses to the reviewers below. Changes were made in the MS Word Track Changes function in the revised manuscript file.

Comment: Authors took into account all of my suggestions. The major concern is the very low number of studies for the meta analysis which explain the high heterogeneity. The results can be considered as negative. There is no impact on outcomes either below 3 minutes or over 5 minutes. This study emphasize that RCTs are needed and open the door to scientific discussion before discussing the current guidelines. The English be evaluated by a native English speaking.

Response: Thank you for your comments. We agreed with the reviewer, as we have discussed this issue in the previous response to reviewer files. We have added a sentence in the conclusion to emphasize the need and opportunity of this topic in our scientific community. (Page 8, Lines 232-233) Also, the current version of this manuscript was reviewed by our native English speaker at our institute.
